# Visible Light Positioning-Based Robot Localization and Navigation

Moi-Tin Chew [1,*] , Fakhrul Alam [2] , Frazer K. Noble [1] , Mathew Legg [1] and Gourab Sen Gupta [1]

1 Department of Mechanical and Electrical Engineering, Massey University, 229 Dairy Flat Highway, Auckland 0632, New Zealand; f.k.noble@massey.ac.nz (F.K.N.); m.legg@massey.ac.nz (M.L.); g.sengupta@massey.ac.nz (G.S.G.)
2 Department of Electrical and Electronic Engineering, Auckland University of Technology, Auckland 1010, New Zealand; fakhrul.alam@aut.ac.nz
* Correspondence: m.t.chew@massey.ac.nz

**Abstract:** Visible light positioning or VLP has been identified as a promising technique for accurate indoor localization utilizing pre-existing lighting infrastructure. Robot navigation is one of the many potential applications of VLP. Recent literature shows a small number of works on robots being controlled by fusing location information acquired via VLP that uses a rolling shutter effect camera as a receiver with other sensor data. This paper, in contrast, reports on the experimental performance of a cartesian robot that was controlled solely by a VLP system using a cheap photodiode-based receiver rigidly attached to the robot's end-effector. The receiver's position was computed using an inverse-Lambertian function for ranging followed by multi-lateration. We developed two novel methods to leverage the VLP as an online navigation system to control the robot. The position acquired from the VLP was used by the algorithms to determine the direction the robot needed to move. The developed algorithms guided the end-effector to move from a starting point to target/destination point(s) in a discrete manner, determined by a pre-determined step size. Our experiments consisted of the robot autonomously repeating straight line-, square- and butterfly-shaped paths multiple times. The results show median errors of 27.16 mm and 26.05 mm and 90 percentile errors of 37.04 mm and 47.48 mm, respectively, for the two methods.

**Keywords:** visible light positioning (VLP); online navigation system; cartesian robot; rolling shutter effect (RSE) camera; spring relaxation (SR); CNC end-effector





## 1. Introduction

Robot navigation [1] is an active topic of research due to the growing demands in sectors like retail services, manufacturing, logistic, healthcare, domestic services, and warehouse management [2]. For many tasks, a mobile robot is required to follow a set path or prescribed route from a start location to destination location or locations. For example, a waiter robot has to go from the kitchen to the customers to deliver food. An autonomously guided vehicle robot needs to traverse between various assembly stations to autonomously deliver parts. Service robots operating in a hospital need navigation to transport food and medication throughout the premises. Cleaning robots need sophisticated techniques to navigate within their service area. The basic parameters of the robot's movement in reaching its destination include speed, positioning accuracy, the type of movement trajectory (e.g., linear), etc. [3]. These are the essentially similar regardless of whether the robot is mobile (e.g., an automated vacuum cleaner) or stationary (e.g., an industrial robotic arm for assembling). These parameters need to be considered while developing a system for controlling a robot.

Outdoor navigation can utilize Global Positioning System (GPS) for robust real-time applications. However, GPS does not function reliably inside buildings due to the degradation of the satellite signals caused by obstructions. Popular methods for robot navigation

in an indoor environment include simultaneous localization and mapping (SLAM), and utilizing the information acquired by its onboard sensors like an odometry sensor, inertia measurement unit (IMU), ultrasonic sensor, electronic compass, light detection and ranging (LiDAR) sensor, and camera [4]. While such techniques are quite mature, robot navigation based on indoor positioning systems (IPSs) has started to get the attention of researchers in recent times [5]. This is driven by the rapid advancement in indoor localization [6] that, apart from robot navigation, can potentially be used for asset tracking, and smart guidance in large facilities like airports. Proprioceptive sensors (IMU, odometry) are subject to cumulative errors [1]. Kidnapped robot or locomotion failure can also create issues as the current location information is lost [7]. Therefore, a global localization-based solution using IPS also offers some inherent advantages by not being susceptible to these issues.

The most common IPSs are based on radio frequency technologies like Wi-Fi [8], BLE [9], ultra-wide band (UWB) [10], and radio frequency identification (RFID) [11]. However, their relatively low localization accuracy means that they are more suited as a secondary source of information for sensor fusion-based navigation [12]. In contrast technologies which offer centimeter-range indoor positioning accuracy such as acoustic signals and visible light signals hold more promise for robotic navigation. Visible light positioning (VLP) has the comparative advantage of being able to leverage pre-existing lighting infrastructure. Light-emitting diodes (LEDs), which are energy efficient, long lasting, and emit low heat, are rapidly becoming the preferred technology for modern indoor lighting infrastructure. The abundance of pre-installed LED lightings presents opportunity to simultaneously use them for visible light communication (VLC) [13] and VLP.

## 2. Related Work

In VLP systems, LED luminaires are used as the transmitting beacons with either a photo diode (PD) or camera as the common receiver sensor attached to the tracked object. Both sensors come with inherent advantages and disadvantages. While cameras may allow for more sophisticated communication schemes [14], PD-based receivers are considerably cheaper and energy efficient [15], and incur less computational cost [16]. Received signal strength (RSS) is the most widely utilized signal characteristic for a PD-based VLP due to simplicity and convenience [17]. VLP that utilizes RSS can either be model- or fingerprinting-based [18]. Fingerprinting-based methods can be time- and labor-consuming compared to the model-based ones. Considering all these points, the VLP system used for this work employed a PD-based receiver that utilizes RSS as the signal characteristic and localizes using a model-based technique. A literature review shows many localization works based on a PD-based receiver utilizing RSS (for example, see the recent survey paper by Rahman et al. [19] on VLP) achieving centimeter-level accuracy. However, there is no reported work that uses such a VLP system as the singular tool for controlling robots in real time.

A survey of recent literature on VLP-based indoor robot navigation yields only a handful of papers. These works utilize a rolling shutter effect (RSE) camera as the light sensing device (please see [20–22] for examples of RSE-based VLP systems). Though the method uses more computer computation time and a complex algorithm because of the involvement of image processing and classification algorithm, the availability of ubiquitous CMOS image sensor cameras in most smartphones has made them the preferred choice among researchers.

Rátosi and Simon [23] employed an RSE camera installed on a mobile robot to determine its pose from the signal received from modulated LED luminaires. While the system is shown to be able to track the moving robot in real time in a 6 m × 6 m room within a low centimeter range, the VLP system is not used for the navigation. The process of decoding LED-ID information from the captured images using an RSE camera can be slow due to the computational latency. Therefore, Li et al. [24] proposed using convolutional neural network (CNN) to increase the speed of identifying the anchor node ID from the captured image, especially with motion blur caused by the fast movement of a robot.

Hua et al. [25] proposed the fusion of a PD- and camera-based system using a version of the Kalman filter. The system is shown to be robust and has the ability to perform real-time positioning. However, the system was not used to control or navigate a robot. Amsters et al. [16] demonstrated the capability of a PD-based VLP that utilizes only unmodulated light. While their system has the potential to estimate the pose of a mobile robot, the experimental results do not show real-time position estimation. They also do not control or navigate the robot using VLP. Zhuang et al. [26] reported on a PD-based VLP system that may be used for the localization of a mobile robot. However, the article only reports localization accuracy; the VLP system is not used to control the movement of the robot.

Guang et al. [5] proposed indoor robot localization based on a VLP system that utilized an RSE camera installed on Turtlebot 3 running a robot operating system (ROS). They presented an extension [27] of this work by introducing a loosely coupled multi-sensor fusion with LiDAR- based SLAM and odometry. Another improvement [28] of the work involves the fusion of VLP and IMU to improve the system's robustness with the ability to handle luminaire shortage/outage. A similar work on the fusion of VLP and IMU to address luminaire shortage/outage issues for an RSE-based receiver can be found in [29]. It should be noted that only one of the articles [28] develops a method for controlling the robot, whereas the other two articles [27,29] focus mainly on the localization aspect.

Therefore, a clear gap in the state of the art can be seen. There is a noticeable lack of works that utilize VLP for controlling a robot. Recent literature shows only a small number of works on robots being controlled by fusing location information acquired via VLP that uses an RSE camera as a receiver with other sensor data. There is no reported work that has used only PD-based VLP to control a robot. This is the motivation of the work being presented in this article.

## 3. Contribution

As far as the authors are aware of, this is the first reported work that utilizes only a PD-based VLP system to control the movement of a robot. In this work, two novel algorithms are proposed and implemented to control the movement of a cartesian robot, constructed in the form of a 2D Computer Numerical Control (CNC) machine, solely by a VLP system in real time. This also allowed for objectively evaluating the efficacy of a PD-based VLP system for controlling a robot. Based on the experimental results collected while the robot is traversing multiple path patterns, both algorithms show promising accuracy.

This is how the rest of the article is organized. Section 4 describes the experimental setup and bespoke hardware. The visible light positioning technique and calibration of the VLP system are discussed in Section 5. The algorithms for controlling the movement of the robot are developed in Section 6. Experimental results are presented and discussed in Section 7. Finally, Section 8 concludes the paper and identifies some potential work that can be carried out to extend the research.

## 4. Experiment Setup

The key equipment of the experimental setup is a purpose-built 2D CNC machine (see Figure 1). The VLP photodiode receiver is mounted on the end-effector. The CNC machine performs three important tasks:

- It acts as the online robotic platform where the end-effector is driven by the VLP system;
- The encoders of the CNC measure the X Y linear movement at the rate of 1000 mm per minute;
- During the online phase, the CNC also records the exact locations of the receiver (end-effector). The location information is not used by the VLP system to drive the CNC, but it is used as the ground truth so that the localization accuracy or error statistics can be computed;
- During the offline phase, the CNC is used to accurately position the VLP receiver at 29 pre-determined locations to collect RSS data for calibrating the Optical Propagation Model of the VLP system.

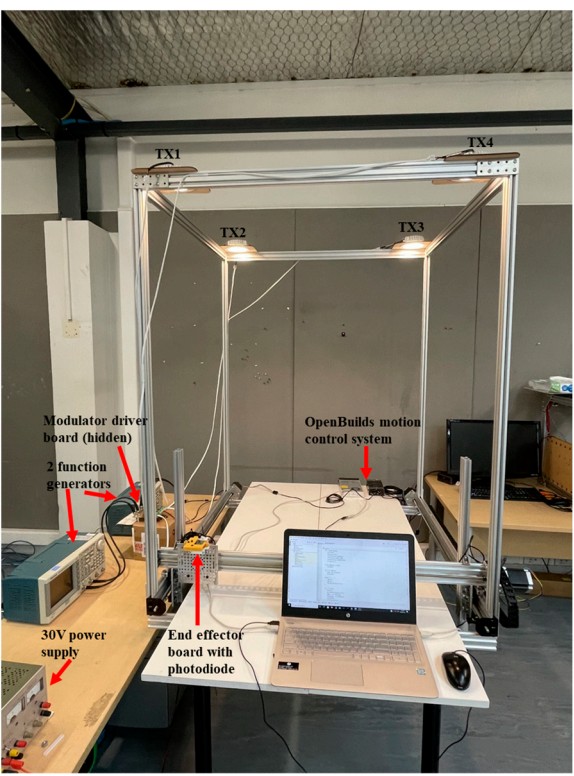

**Figure 1.** VLP system test rig.

The end-effector's minimum step size is 0.1 mm, achieved through lead screw actuation and Nema 23 stepper motors. An OpenBuilds BlackBox Motion Control System (MCS) with Grbl firmware controls the stepper motors. The host PC sends commands to the BlackBox MCS over a USB, with desired positions converted into GCODE commands via a Python program. These desired positions were computed by the VLP.

The VLP system consists of four transmitters constructed with LED luminaires which are commercially available and can be bought easily off the shelf. These luminaires are placed at a 1050 mm height above the CNC at known x–y locations: TX1 (0, 0), TX2 (0, 780), TX3 (760, 780), and TX4 (760, 0) (see Figure 1).

For the experiments conducted, an unmodulated sine wave of frequencies of 2 kHz, 2.6 kHz, 3.2 kHz, and 4.4 kHz were inserted by a modulator driver circuit powered by a 30 V power supply (shown in Figure 2). The input of the modulator circuit is connected to a function generator which supplies the required sine wave. Each function generator has two signal generation channels. Altogether, two function generators are sufficient for this experiment (as seen in Figure 1).

Underneath, the PD-based VLP receiver was mounted on the end-effector of the CNC. Together they form the robot that is controlled by utilizing the location estimated with respect to the fixed luminaires. On the receiver board, the photodiode sensor converted the optical signal to an electrical signal which was then amplified through a transimpedance amplifier before being processed by an onboard analog to digital convertor (ADC). More details of the receiver board can be found in [30]. The sensor board receives and decodes the identities of these four lights using a Python program in the PC. After that, the data are transmitted out serially via USB to PC for Fast Fourier Transform (FFT) processing. The function of the FFT is to demultiplex and measure the magnitude of the four sinewaves present in the received signal [30]. The magnitude squares constitute the RSS set at each location. Please refer to Figure 3 for more details of the operation of the VLP system.

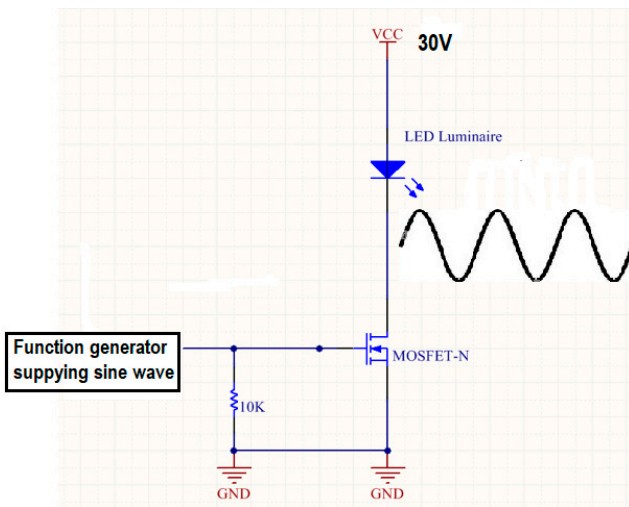

**Figure 2.** Modulator driver circuit.

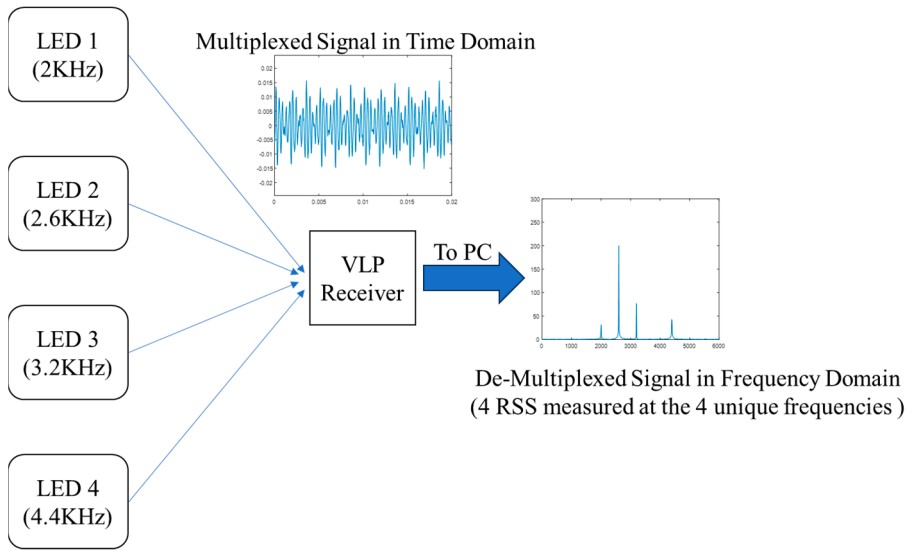

**Figure 3.** Key concept of the VLP system. RSS is extracted from the received signal using FFT. A representative RSS from four visible luminaires is shown at the receiver at a particular location. The measured RSS is used for model calibration during the offline stage and for ranging and localization during the live stage.

## 5. Localization Using VLP

During the offline stage, the RSS–distance relationship is found by calibrating the Lambertian propagation model. RSS data are collected at a set number of pre-determined locations for the parameter calibration. During the online phase, the live RSS readings are used to find the distance of the target device (the receiver on the end-effector) from each transmitter by "inverting" this calibrated propagation model. Subsequently, the receiver's position is estimated via lateration.

### 5.1. Offline Calibration of VLP

For this research, the receiver/target plane and luminaire/transmitter plane are kept horizontal with respect to one another. This allows for the simplification of the Lambertian propagation model [18]. Please refer to the discussion in Section 8 on how this limitation can be addressed in the future.

Under this assumption of a parallel arrangement, the received power, $P_{r_i}$, or the RSS at a distance $d_i$ from the $i$th luminaire (Figure 4) within the field of view of the luminaire

can be simplified from the Lambertian propagation model following the process outlined in [18].

$$P_{r_i} = P_{r_{i,0}} \left( \frac{d_{i,0}}{d_i} \right)^{m_i+3} \tag{1}$$

**Figure 4.** Parameters of the Lambertian propagation model.

Here, $m_i$ is the Lambertian order and $P_{r_{i,0}}$ is the RSS at a *reference* location at a distance $d_{i,0}$ from the target. Equation (1) can be rearranged so that

$$m_i = \left[ \frac{log\left( \frac{P_{r_i}}{P_{r_{i,0}}} \right)}{log\left( \frac{d_{i,0}}{d_i} \right)} \right] - 3 \tag{2}$$

Figure 5 shows the position of the 29 offline locations where the RSS values were collected to estimate $m_i$ for calibrating the RSS–distance relationship. As suggested by [31], the reference location is chosen directly underneath the corresponding luminaire. The Lambertian order is estimated at the rest of the 28 locations using Equation (2), and then averaged over these 28 values. These offline locations are carefully chosen so that all regions of the Lambertian propagation models for each luminaire are captured [32]. Table 1 shows the estimated Lambertian order and Figure 6 shows the measured RSS values and the calibrated RSS–distance Lambertian curves of the four luminaires.

**Table 1.** Estimated Lambertian order of the luminaires.

| Luminaire | Lambertian Order ($m_i$) |
|---|---|
| 1 | 4.60 |
| 2 | 3.98 |
| 3 | 3.20 |
| 4 | 3.58 |

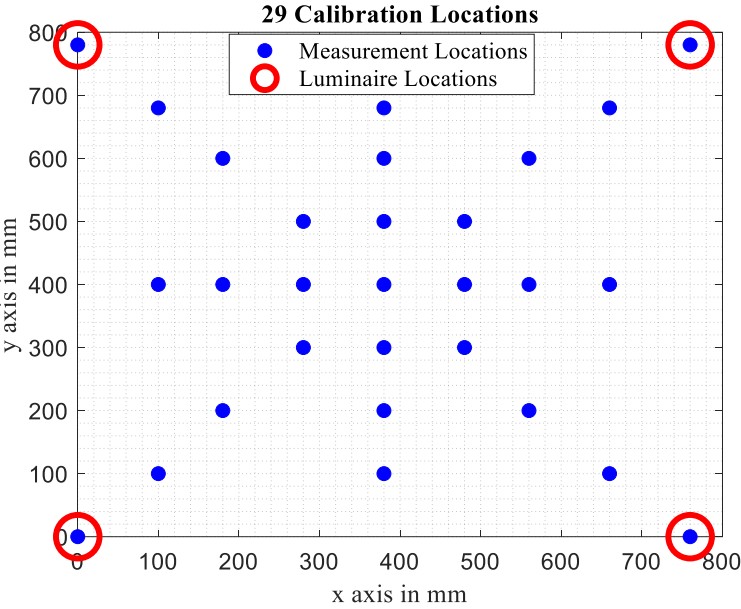

**Figure 5.** The location of the luminaires and the 29 measurement locations for offline parameter calibration of the Lambertian propagation model.

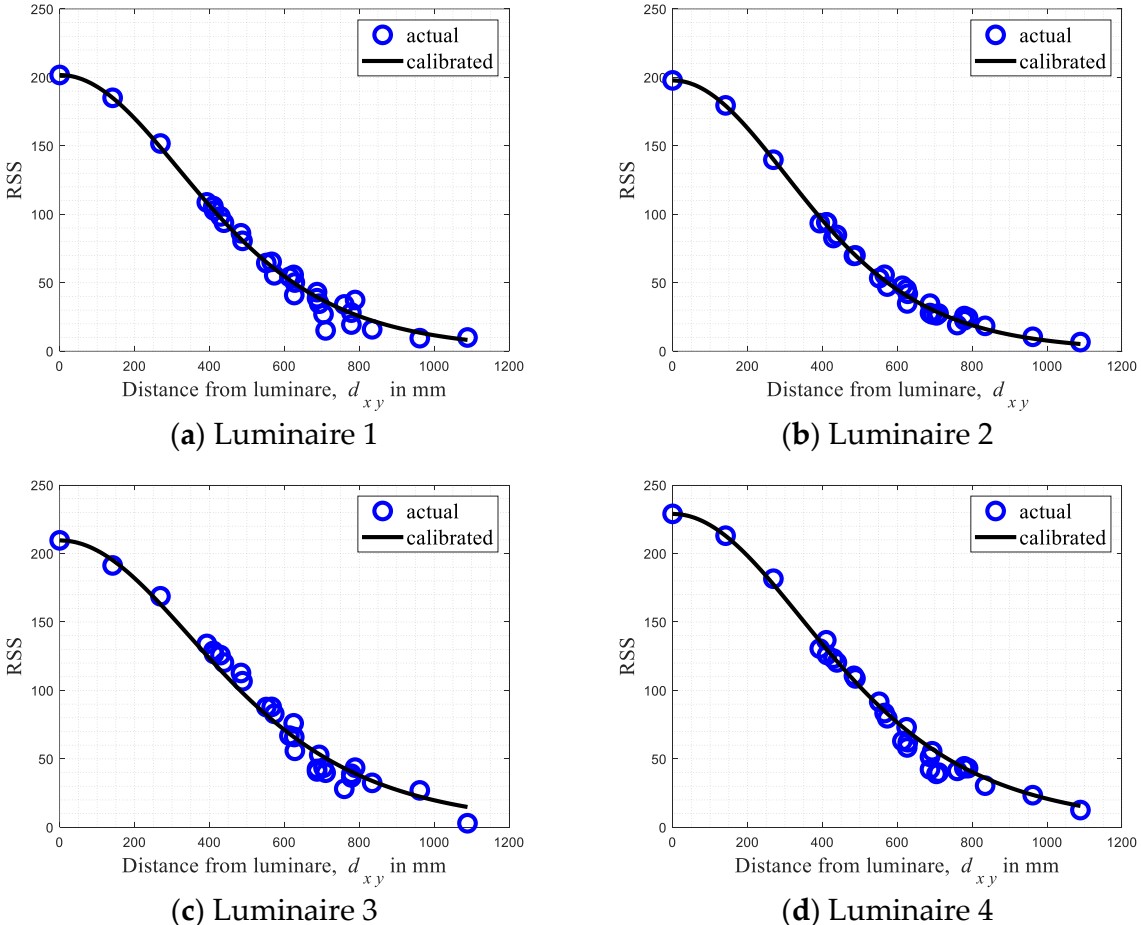

**Figure 6.** Calibration of the Lambertian propagation models for the four luminaires.

*5.2. Online Localization Using VLP*

Once the RSS–distance model is calibrated for each luminaire, Equation (1) can be rearranged for ranging (i.e., to compute the distance of the target from a luminaire) so that

$$d_i = d_{i,0} \left( \frac{P_{r_{i,0}}}{P_{r_i}} \right)^{1/(m_i+3)}, \tag{3}$$

The horizontal distance of the target from the *i*th luminaire, $d_{xyi}$ (please see Figure 4) is computed as

$$d_{xyi} = \sqrt{d_i^2 - h^2} \tag{4}$$

## 6. Online Navigation

We developed two algorithms that can guide the end-effector to move from a starting point to the target/destination point(s). Both algorithms drive and make the end-effector move in a discrete manner, determined by the step size.

At each iteration of the algorithm, (1) visible light signal is captured by the VLP receiver (for 500 ms) and sent to the PC for extraction of RSS, (2) the distance of the end-effector from each luminaire is determined (as per the procedure discussed in the previous section), and (3) the end-effector is instructed to move by a step size toward a direction as determined by the algorithm or to remain stationary if the destination is reached. To maneuver the CNC machine, a GCODE set is needed to be sent to the OpenBuilds BlackBox motion control driver. The main program which is written in Python works out the x and y direction step movement from the algorithm and generates the equivalent steps in GCODE. The Black box motion control driver then processes the given GCODE and commands the movement of the stepper motors connected to the CNC machine.

*6.1. Algorithm 1*

The flowchart in Figure 7 shows the algorithm. The robot, if required, moves at a step size of *T*. However, before any movement takes place,

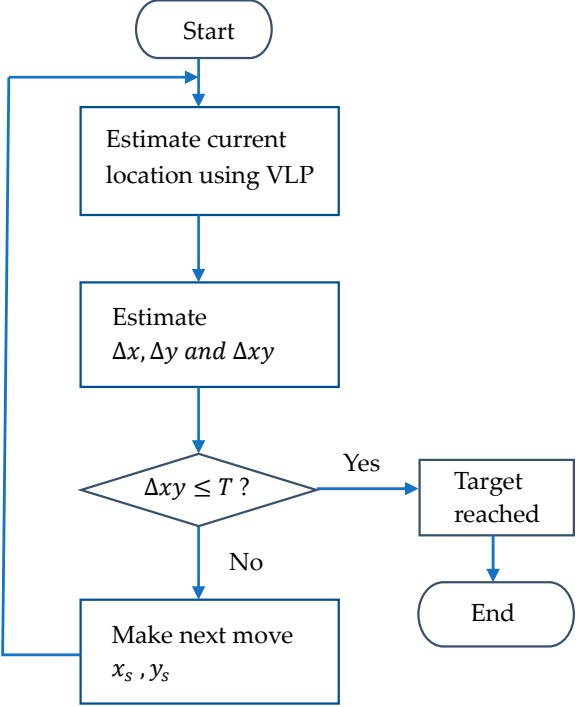

**Figure 7.** Flowchart of Algorithm 1. Please see Figure 8 for details on $\Delta x$, $\Delta y$, and $\Delta xy$.

(i)    The robot's current distance from each luminaire's position is determined using live VLP reading following the process outlined in Section 5.2. Once the horizontal distance of the target from all four luminaires is found, the location of the target on the xy plane is estimated using multi-lateration [24].

(ii)    The direct distance between the current position and the final destination position (refer to Figure 8), $\Delta xy$, is calculated. $\Delta xy$ is based on the difference in x distance, $\Delta x$, and the difference in y distance, which is $\Delta y$, so that $\Delta xy = \sqrt{\Delta x^2 + \Delta y^2}$.

(iii)    If $\Delta xy \leq T$, the robot is considered to have reached the target, otherwise (i.e., if $\Delta xy > T$) the robot receiver has not reached the destination yet and moves to a new position by moving $x_s$ and $y_s$ in the x and the y directions, respectively. Here $x_s = \frac{\Delta x}{\Delta xy} \times T$ and $y_s = \frac{\Delta y}{\Delta xy} \times T$. As discussed previously, $T$ is the step size (which is 20 mm for our experiments). It should be noted that

    1.    if $x_s = +ve$, the robot moves to the right;
    2.    if $x_s = -ve$, the robot moves to the left;
    3.    if $y_s = +ve$, the robot moves up;
    4.    if $y_s = -ve$, the robot moves down.

Once arriving at the new position, steps (i)–(iii) are reiterated until the robot arrives at its destination.

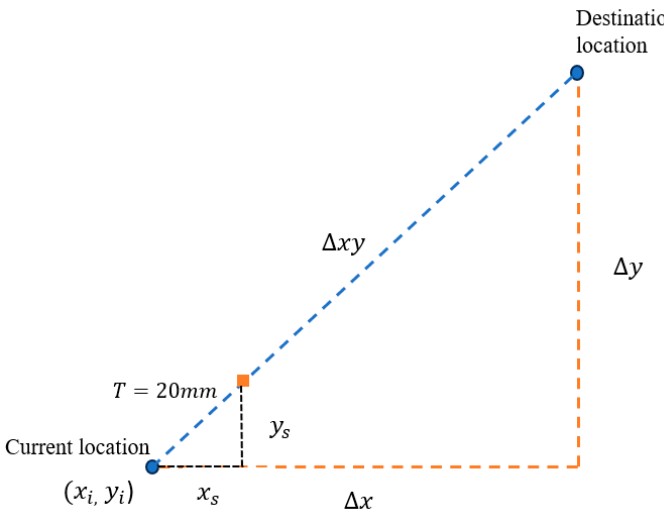

**Figure 8.** Robot travels in a planned straight path from the current location to the destination location.

### 6.2. Algorithm 2

This algorithm is based on the spring relaxation [30] or artificial potential field [33] techniques. In this section, we explain the algorithm using the concept of spring relaxation which uses fictitious springs and Hooke's law to guide the robot to its destination/s. Figure 9 illustrates the concept of spring relaxation.

The location of the luminaires is the anchor to which the one end of each fictitious spring is connected. The other ends are connected to the robot. The distance between the target destination and each anchor is the natural length of each corresponding spring. Therefore, when the robot is at its destination, all the springs are relaxed (neither compressed nor stretched) and no force is acting on any of them. The goal of the algorithm is to move the robot to a position where the net force acting on the springs is zero (or below a certain threshold). Figure 10 shows the flowchart of the algorithm.

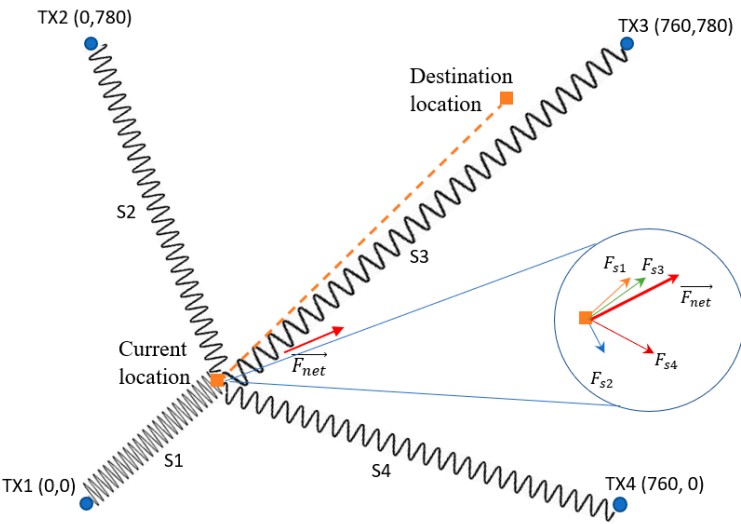

**Figure 9.** Spring relaxation technique to guide CNC robot sensor to travel from current location to the destination location; it shows the various forces exerted by each spring, $F_{s1}$ to $F_{s4}$, and the resultant net force, $\overrightarrow{F_{net}}$.

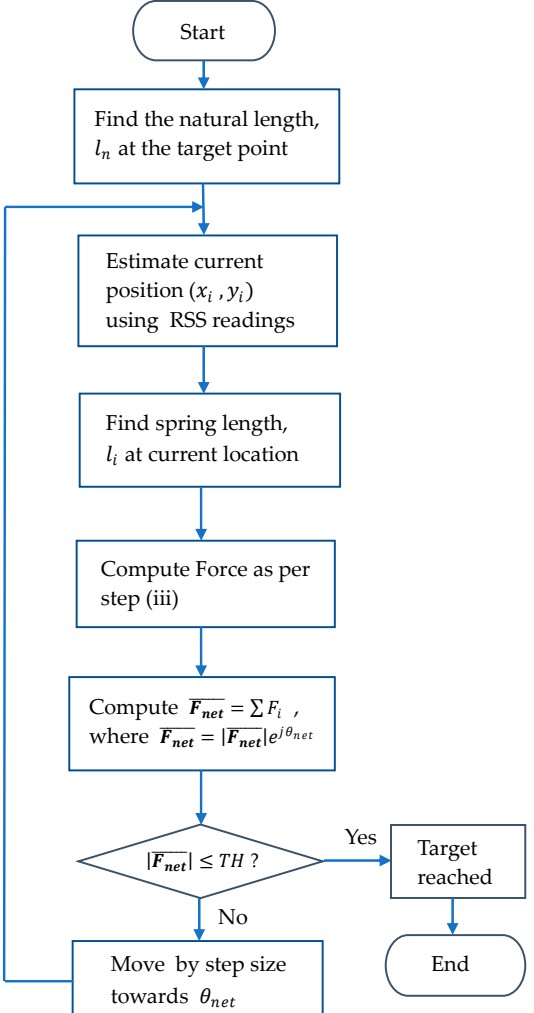

**Figure 10.** Flow chart of Algorithm 2.

(i)     Compute the natural length, $l_{n,i}$ ($i = 1$ to 4 for our experimental setup), of the four fictitious springs of the sensor device at the target position.

(ii)    Take an RSS reading at the current position, $x_i, y_i$. Next, calculate the distance of the robot from each luminaire using the ranging process described in Section 5.2 Equations (3) and (4). Each of these distances represents the current length of each spring, $l_i$ ($i = 1$ to 4 for our experimental setup).

(iii)   If the current location is not the destination, the springs are not in an equilibrium state (either compressed or stretched). A compressed spring experiences a push away from the anchor, whereas a stretched spring experiences a pull toward the anchor. The individual force, $F_i$, exerted by each spring, is estimated as $F_i = |l_{n,i} - l_i|e^{j\theta_{li}}$, where $|l_n - l_i|$ is the magnitude of the force and $\theta_{li}$ is the direction of the corresponding force.

(iv)    Compute the net force acting on the robot as $\overline{F_{net}} = \sum F_i$. Note that $\overline{F_{net}} = |\overline{F_{net}}|e^{j\theta_{net}}$, where $|\overline{F_{net}}|$ is the magnitude of the net force and $\theta_{net}$ is the direction of the net force. If the net force is smaller than or equal to a threshold ($|\overline{F_{net}}| \leq TH$), the robot is at the target. If not (i.e., $|\overline{F_{net}}| > TH$), the robot moves toward the direction of the net force at the step size $T$. Steps (i)–(iv) are repeated and the robot moves until it reaches the destination.

Figure 11 shows the tasks completed by the PC to perform the navigation of the end-effector from the starting point to the next location.

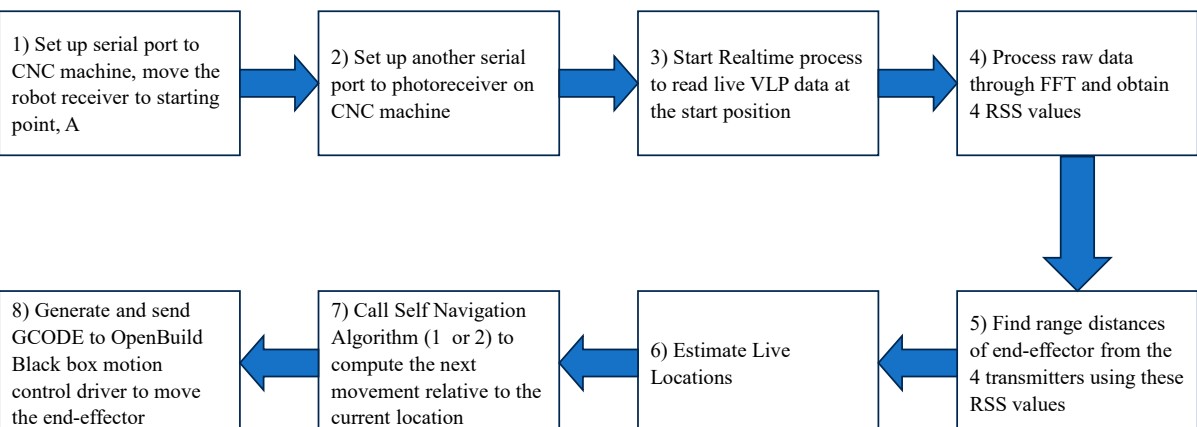

**Figure 11.** The tasks completed by the PC to control the movement of the CNC end-effector while guided by one of the two algorithms running on live RSS data. Steps 3 to 8 are iterated until the end-effector reaches the destination (when the loop ends after step 6).

## 7. Experimental Results

Figures 12–14 show how the CNC end-effector traverses through various paths. Each path is repeated five times. The actual location of the end-effector can be determined to an accuracy of 0.1 mm since the CNC tracks and records its position. Please note that this information is only used as the ground truth and is not used to control the end-effector which is solely driven by one of the algorithms based on the acquired RSS. As can be seen, VLP-driven algorithms can control the robot reasonably accurately (accuracy statistics are discussed later). However, it is also clear that for all patterns, the robot does deviate from the ideal straight-line path. This is because the VLP-based position estimate using RSS data has some error. Due to this and the discrete nature of the robot's movement (step size of 20 mm), it does not reach the "target" points precisely. For every pattern, each path traversed is also slightly different regardless of the algorithm utilized. Therefore, the robot and the VLP-based control system, in its current state, does not have the precision or repeatability to be used for applications like assembly of electronic devices [34].

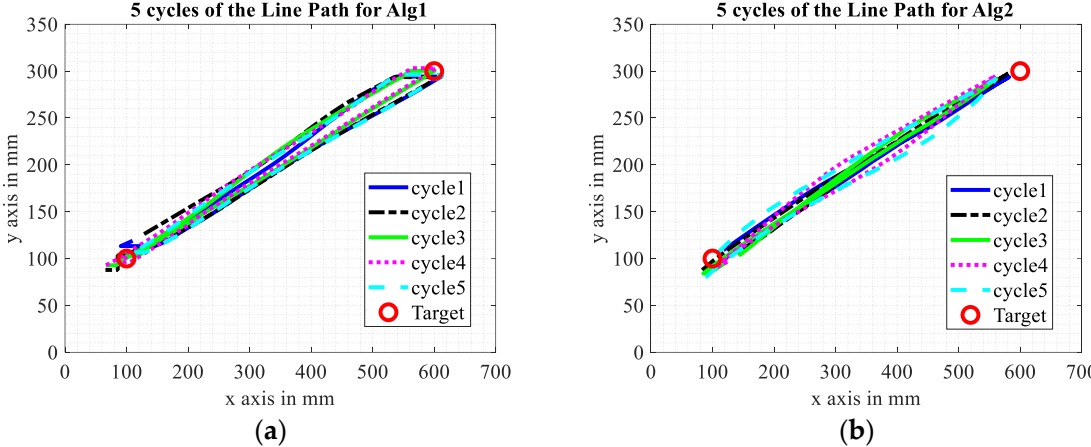

**Figure 12.** "Line Path" travelled by the robot. The robot is moving from point A (100, 100) to point B (600, 300) and then returning back to A. The diagrams ((**a**,**b**) for algorithms 1 and 2, respectively) show five iterations of the path.

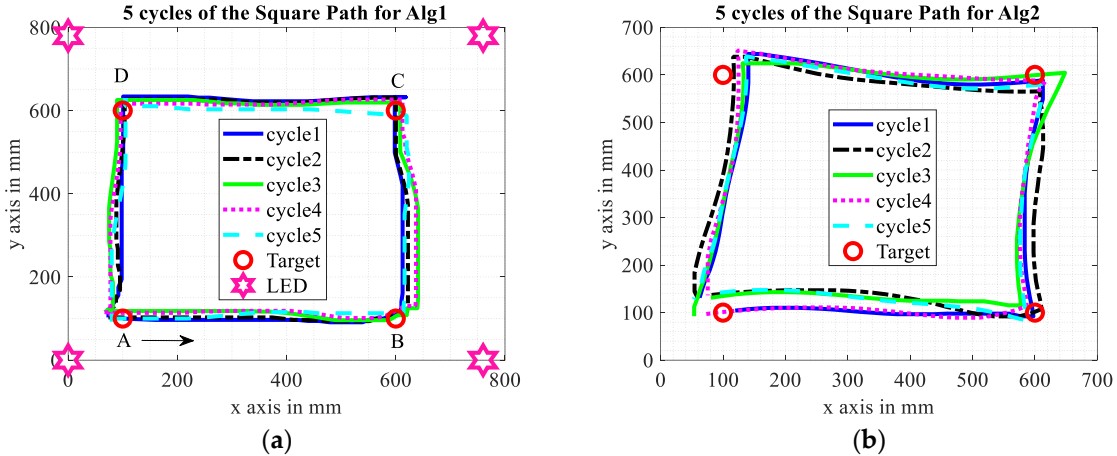

**Figure 13.** "Square Path" travelled by the robot. The robot is moving from point A (100, 100) to point B (600, 100) to point C (600, 600) to point D (100, 600) to point A. The diagrams ((**a**,**b**) for Algorithms 1 and 2, respectively) show five consecutive iterations of the path.

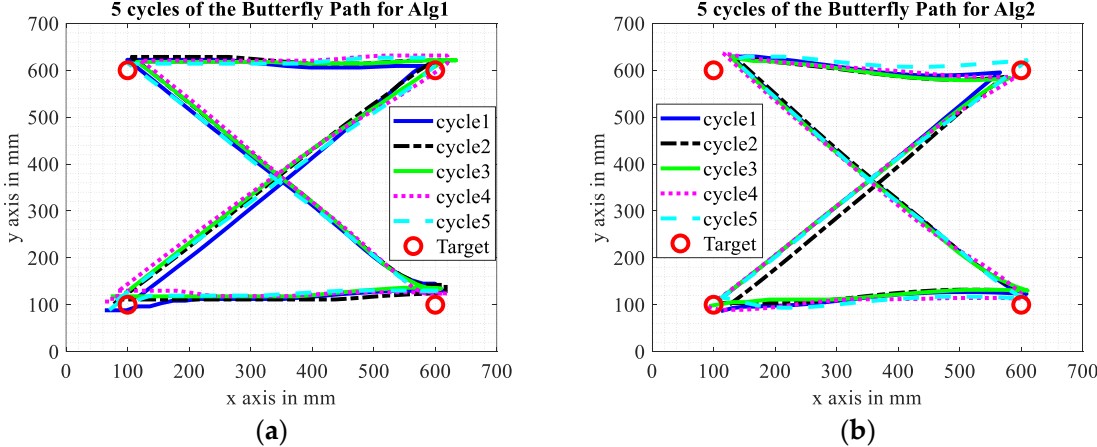

**Figure 14.** "Butterfly Path" travelled by the robot. The robot is moving from point A (100, 100) to point C (600, 600); then point C to point D (100, 600); then point D to B (600, 100) and then point B to A. The diagrams ((**a**,**b**) for Algorithms 1 and 2, respectively) show five iterations of the path.

Figure 15 shows the cumulative distribution function (CDF) of the errors for each path. The error is computed as the amount of the deviation of the robot's locations from an ideal straight line. Figure 16 shows the CDF of the errors in reaching the target (measured as the difference between the destination point and where the robot stopped) for all the paths. The median and 90-percentile errors are shown in Table 2. It should be noted that the errors in reaching the target can be reduced by lowering the step size (e.g., 27.16 mm to 15.23 mm for Alg1 when the step size is changed from 20 mm to 5 mm). But that comes at the cost of higher computational cost and increased time.

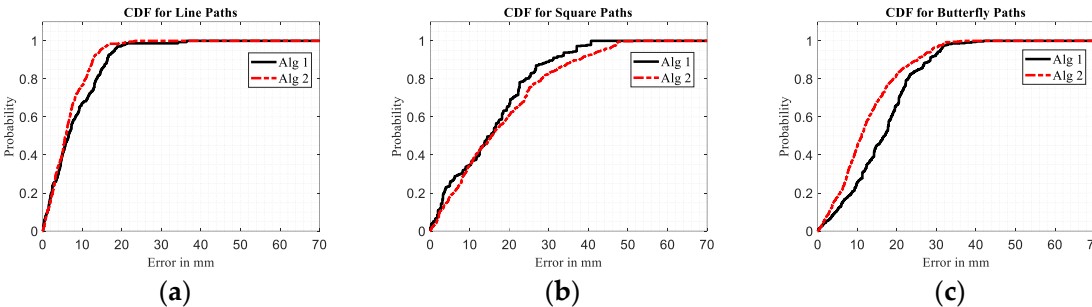

**Figure 15.** CDF showing the accuracy of both algorithms for (**a**) Line Paths, (**b**) Square Paths, and (**c**) Butterfly Paths.

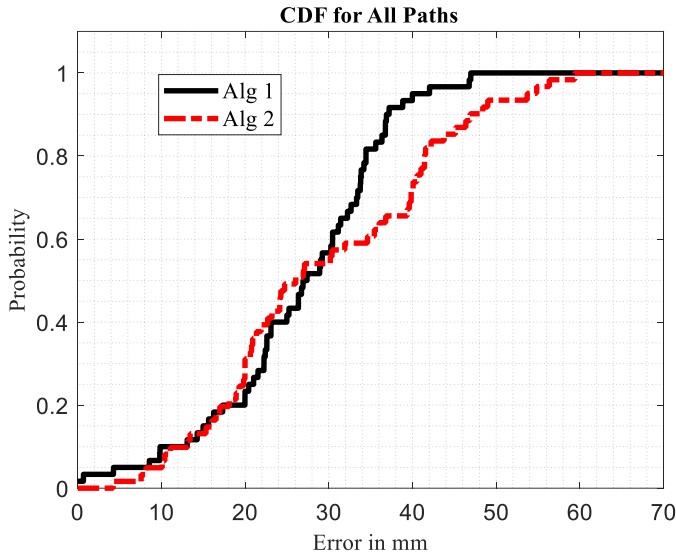

**Figure 16.** Comparing "Target" positions accuracy.

**Table 2.** Accuracy statistics for both algorithms for various scenarios.

| Scenario | Median Error (mm) | | 90-Percentile Error (mm) | |
|---|---|---|---|---|
| | **Algorithm 1** | **Algorithm 2** | **Algorithm 1** | **Algorithm 2** |
| Line Path | 8.02 | **5.86** | 16.34 | **12.71** |
| Square Path | **15.15** | 15.51 | **30.99** | 37.59 |
| Butterfly Path | 16.81 | **11.1** | 28.48 | **25.19** |
| Destination Points | 27.16 | **26.05** | **37.04** | 47.48 |

## 8. Conclusions and Future Works

This work presented the control of a cartesian robot, a 2D CNC machine, capable of precise positioning along the X and Y axes using only a PD-based VLP system. Two algorithms were developed to control the position of the robot in real time. Both algorithms

show promising accuracy. Algorithm 1 performed better for the "square path", achieving a 15.15 mm median localization error. Algorithm 2 performed better for the "line path" and the "butterfly path", achieving median localization errors of 5.86 mm and 11.1 mm, respectively. Given the accurate performance of the system, further investigation is warranted. Also, further investigation needs to be carried out before an algorithm can be conclusively recommended. There are several other limitations that can be addressed in the future.

The locations of the luminaires of the VLP have not been optimized. Future works can explore the optimization of the layout of the VLP system. However, it should be noted that as part of real-world implementation, the location of the luminaire may be dictated by the need for illumination.

The Lambertian model was simplified under the assumption that the PD and the luminaires are parallel. If the receiver tilts, the localization estimate is affected. However, there are models [35] that address this issue and can thus be adopted. A simple gimbal can also ensure that the receiver maintains its orientation.

Pose comprises of location and orientation. For this work, the location was determined via VLP. The orientation was determined via a CNC. Therefore, developing a new receiver capable of VLP-based orientation estimation will enable more sophisticated control strategies. Bernades et al. [36] proposed a multi-PD based system for estimating the orientation of the receiver for an infrared (IR)-based positioning system. A similar approach can be applied for estimating the complete pose (both location and orientation) of the receiver. Incorporating orientation information into the algorithm and conducting new benchmarking experiments will open new avenues for investigation. Obstacle avoidance and path planning can also be investigated in the future. But this will require utilizing other sensors alongside PD.

**Author Contributions:** Conceptualization, M.-T.C. and F.A.; methodology, M.-T.C. and F.A.; software, M.-T.C., F.A. and M.L.; validation, F.A.; formal analysis, M.-T.C. and F.A.; investigation, M.-T.C.; resources, F.K.N. and F.A.; data curation, M.-T.C., F.A. and M.L.; writing—original draft preparation, M.-T.C.; writing—review and editing, F.A. and F.K.N.; visualization, M.-T.C. and F.A.; supervision, F.A., M.L. and G.S.G.; project administration, M.-T.C.; funding acquisition, F.A. All authors have read and agreed to the published version of the manuscript.

**Funding:** This research received no external funding.

**Data Availability Statement:** The data presented in this work are available upon request from other researchers and authors.

**Acknowledgments:** The authors wish to thank Alex Xu for his help in setting up the VLP experiment and data collection.

**Conflicts of Interest:** The authors declare no conflicts of interest.

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
