# Peer review of "Visible Light Positioning-Based Robot Localization and Navigation"

_electronics, doi:10.3390/electronics13020368_

Round 1

Reviewer 1 Report

Comments and Suggestions for Authors

This paper investigates the right-angle robot control method by VLP system only and designs experiments to verify to get its effectiveness. In my opinion, this paper is publishable.

1.      Is there a basis for the layout of the four LEDs in the experimental design of Part IV?

2.      The paper describes Algorithm 1 and Algorithm 2 in the "Line Path", "Square Path", "Butterfly Path" in different paths in the error is different, in the practical application of the two algorithms, how to choose?

3.      The RSS data corresponding to 4 LEDs were obtained in the experimental session, can the purpose of the experiment be accomplished with 3 LEDs?

Author Response

Dear Reviewer 1 colleague,

We appreciate your valuable comments, and we hope that we have addressed your concern/query on our paper- please see the attached file. Thank you.

Reviewer 2 Report

Comments and Suggestions for Authors

This paper mainly studies the robot localization and navigation technology based on visible light positioning. Two algorithms are proposed and implemented to control the movement of a cartesian robot, constructed in the form of a 2D CNC machine, solely by a VLP system in real time. The following comments are suggested to further improve it:

1.     Please rewrite the Abstract. The abstract should be a brief statement of the content of paper. The abstract of this manuscript describes the experimental device of the Cartesian robot in a large space, and does not specifically describe the two control methods.

2.     The authors should increase their efforts to improve their contributions and should further strengthen the motivation for the preparation of this manuscript. The Contribution section should list the innovation points.

3.     In this manuscript, there are too many words to describe and lack some expressions of mathematical formulas.

4.     Please check the algorithm flowchart carefully and draw according to the standard specification of the flowchart.

5.     In order to ensure beauty, the fonts in Figure 3 should be unified as much as possible.

6.     Please examine the formatting of the full text carefully, such as the lack of space between line 104 “[26-28]” and “focus”. The 2 in Equation 2 on line 200 should be parenthesized.

7.     In algorithm 1, the distance between the robot and the target point is greater than T and less than T. If it is equal to T, how to deal with it?

8.     In order to make the path clearer, the ordinate of Figure 12 can be reduced appropriately.

9.     The second paragraph of the Conclusion is very abrupt. Should this part be described in the relevant work in Section 2?

Comments on the Quality of English Language

Please improve the quality of the English language by following the English presentation required by Energies.

Author Response

Dear Reviewer 2 colleague,

We appreciate your valuable comments on our manuscript. We hope that we have addressed your concern/query on our paper - please see attached file

Reviewer 3 Report

Comments and Suggestions for Authors

This manuscript presents the control of a cartesian robot, a 2D CNC machine, capable of precise positioning along the X and Y axes using only a PD-based VLP system, and two algorithms were developed to control the position of the robot in real time. It’s interesting.

After a careful review of the manuscript, I am suggesting the following comments to the authors to revise the manuscript.

1.      “CNC” appears many times in this manuscript, it is recommended to indicate the full name when it appears for the first time.

2.      What is the difference between “a photodiode based VLP” in line 111 and “a PD-based VLP” in line 112? If there is no difference, it is recommended to revise.

3.      Line 189: If Equation 1 refers to other literature, it needs to be marked.

Author Response

Dear Reviewer 3 colleague,

We appreciate your valuable comments on our manuscript. We hope that we have addressed your concern/query on our paper- please see the attached file. Thank you.

Reviewer 4 Report

Comments and Suggestions for Authors

Dear Authors,

please consider the following comments to improve your manuscript:

1. Lines 27-29 - you have provided general areas for robot navigation. It would be interesting to also add specific robot navigation applications in these areas.

2. Line 29-30 - You wrote that the robot must follow a path or designated route from the starting location to the destination location. Yes, but apart from the description of navigation methods later in the text, there is no basis here related to the basic parameters of the robot's movement to reach the target position.

The parameters will be similar for both mobile and stationary robots.

At this point, please add an additional paragraph covering this content.
Please refer to the literature:

- This work presents a description of the basic parameters of the robot's movement, such as speed, positioning accuracy, and the type of movement trajectory

https://doi.org/10.1016/j.jmapro.2023.06.063

- next, please describe the precision of robots' movement

https://doi.org/10.1016/j.procir.2020.05.247

3. Line 122 - please provide model of the CNC machine

4. "(see Figure 4)" -Please remove "see".

5. Table 1 - hundredths seem to be a better way to present numerical data in this case. 4.60, 3.98, etc.

6. Fig. 6 - what are the units, especially distances?

Author Response

Dear Reviewer 4 colleague,

We appreciate your valuable comments on our manuscript. We hope that we have addressed your concern/query on our paper - please see the attached file. Thank you.

Round 2

Reviewer 2 Report

Comments and Suggestions for Authors

There are still format problems in the revised manuscript that require further improvement, although the author has already responded to some of our questions.

1.The author should continue to modify the algorithm flowchart. The standard flowchart should start with the 'Start' symbol and end with the 'End' symbol. And the arrow of the loop path should not be on the process box.

2.The font size of the text in figures should be smaller than the text.

Comments on the Quality of English Language

There are still format problems in the revised manuscript that require further improvement.

Author Response

Dear Reviewer 2 colleague,

Thank you again for your further comments on our manuscript! We hope that we have addressed your questions as pointed out as below :

1.The author should continue to modify the algorithm flowchart. The standard flowchart should start with the 'Start' symbol and end with the 'End' symbol. And the arrow of the loop path should not be on the process box.

 Answer: The two flowcharts have been further modified.

 2.The font size of the text in figures should be smaller than the text.

Answer: Figures 5 and 16 have been updated